# Imported Typhoid Fever in Romania Between 2010 and 2024

**DOI:** 10.3390/idr17020016

**Published:** 2025-02-25

**Authors:** Dragos Stefan Lazar, George Sebastian Gherlan, Simin Aysel Florescu, Corneliu Petru Popescu, Maria Nica

**Affiliations:** 1Infectious Diseases Department, Carol Davila University of Medicine and Pharmacy, 050474 Bucharest, Romania; dragos.lazar@umfcd.ro (D.S.L.); simin.florescu@umfcd.ro (S.A.F.); 2Infectious Diseases Department, Dr. Victor Babes Clinical Hospital for Infectious and Tropical Diseases, 030303 Bucharest, Romania; corneliu.popescu@umfcd.ro (C.P.P.); maria.nica@spitalulbabes.ro (M.N.); 3Virology Department, Carol Davila University of Medicine and Pharmacy, 050474 Bucharest, Romania; 4Microbiology Department, Carol Davila University of Medicine and Pharmacy, 050474 Bucharest, Romania

**Keywords:** typhoid fever, *S.* Typhi, aneosinophilia, tachycardia, multi-drug resistance

## Abstract

Background/Objectives: Although a “forgotten” disease in developed countries, typhoid fever remains a significant global health problem, especially in regions with inadequate sanitation and overcrowding. Despite medical advances, this systemic bacterial infection, caused by *Salmonella* Typhi, continues to affect millions worldwide. Accurate diagnosis and timely treatment are crucial to prevent severe complications and mortality. Even though antibiotic therapy is effective, the emergence of drug-resistant strains is a growing challenge. Methods: We present a series of cases encountered in a tertiary infectious disease hospital in Romania over 15 years. Results: The hospitalised patients were mainly from Sub-Saharan Africa and the Indian subcontinent; the median time between the onset of the first symptoms and hospital admission was 15 days. The symptoms encountered along with fever were headache, chills, cough, diarrhoea and tachycardia, an unusual feature in the clinical picture of this disease. Aneosinophilia (the absence of peripheral eosinophilic granulocytes) was the most frequently encountered laboratory finding, followed by increased serum transaminases and inflammatory syndrome. Conclusions: *S*. Typhi was generally identified from blood culture, demonstrating, except in one case, resistance to ciprofloxacin and, in several cases, multi-drug resistance (MDR). In this series of cases, all strains were sensitive to ceftriaxone.

## 1. Introduction

Despite advances in research and prevention strategies, typhoid fever continues to be a public health problem in various countries. This is an example of a disease that divides the human population of the 21st century into two parts: those in countries with urban development, sanitation, and broad access to clean water sources, where the incidence is extremely low, and those in overpopulated and poor regions of the world, including Asia, Africa and South America, where there are significant levels of incidence. The aetiological agent is *Salmonella enterica* serovar Typhi (*S.* Typhi). *S. Paratyphi* A, B and C are the other serotypes of *Salmonella* that can cause a similar disease, namely paratyphoid fever. It is estimated that 11–21 million cases of typhoid fever and 5 million cases of paratyphoid fever occur worldwide each year, causing approximately 135,000–230,000 deaths [1]. The cases that appear in European countries, Australia and North America are mostly “imported” cases. The most exposed population in these countries is the one that travels to endemic areas to visit relatives or for profit-making purposes, with organised tourism accounting for a small number of these cases. For example, in the European region, the United Kingdom (UK) and France have a higher incidence of cases than other countries in the region. Thus, the last European Centre for Disease Prevention and Control (ECDC) report that included both countries (2019) indicated 279 cases in France and 533 in the UK [2].

Historically, the disease was confused for many centuries with other febrile diseases accompanied by typhoid fever. Gr. “Typhoid”, which means “smoke”, was used to describe the delirium that patients presented with this disease. Although several authors described the disease in previous centuries (Peyer in 1677, Willis in 1684 and Huxham in 1739), at the beginning of the 19th century, the disease was clinically and anatomopathologically differentiated by Louis (1819), who called it *fièvre typhoide*, and by Ritchie (1847), who called it enteric fever. The causative agent was discovered by Eberth in 1880, and, in 1896, Widal introduced the seroagglutination reaction, a diagnostic method whose principles are still used today [3].

The second half of the 19th century represented the emergence of significant epidemics associated with numerous armed conflicts. For example, during the American Civil War (1861–1865), over 75,000 cases of typhoid fever were recorded, with a mortality rate of 36% among Union soldiers [4], and, during the Russo–Ottoman War of 1877–1878, in the city of Philippopolis (now Plovdiv), the incidence exceeded 10% [5]. The same incidence rate (10%) was reported among the Romanian army involved in this war, with the highest incidence ever reported in Romania [6]. The 20th century came with a progressive decrease in disease incidence in Europe and North America as sanitation and water quality improved. With the advent of antibiotics and effective vaccination, the incidence rates have decreased dramatically worldwide.

With the start of effective antibiotic treatment, bacterial resistance has also emerged. Thus, resistance to first-line antimicrobials, such as chloramphenicol, trimethoprim-sulfamethoxazole (TMP-SMX) and ampicillin (AMP), including multi-drug resistance (MDR) [7] and non-susceptibility to at least one antimicrobial agent in three or more antimicrobial classes), was first identified in the 1970s and became common in the early 1990s. Resistance to ciprofloxacin (CIP) was reported in the mid-1990s and became consistently encountered after 2000, and, in 2016, resistance to third-generation cephalosporins (ceftriaxone) was also reported [8]. MDR strains that acquired resistance to fluoroquinolones and ceftriaxone (CTX) were considered to have extensive drug resistance (XDR [7]—non-susceptibility to at least one antimicrobial agent in all but two or fewer antimicrobial classes). It is a worrying finding that, currently, azithromycin (AZM) remains the only effective oral antibiotic that is useful in treating non-hospitalised patients in Southeast Asia. However, even in this situation, isolated cases of resistance have been reported in Nepal, Bangladesh and India [9,10,11].

In countries with adequate sanitation, typhoid fever has become, for the general public and many professionals, a disease of mostly historical interest or a disease that is “foreign” to these countries. The historical success seen in controlling typhoid fever in high-income countries has not been reported in low- and middle-income countries, where access to clean water and the poor sanitation and healthcare infrastructure have contributed to the emergence of strains with increasingly extensive antibiotic resistance [12].

In this paper, we present a series of cases of typhoid fever that were diagnosed and/or treated in a tertiary infectious disease hospital. We seek to return the focus to a disease that, despite now having an exceptionally low incidence due to its history and re-emergence capacity, could become a public health problem at any time.

## 2. Materials and Methods

We evaluated cases accepted at the Dr. Victor Babes Hospital for Infectious and Tropical Diseases (VBH) in Bucharest, Romania, diagnosed with typhoid fever according to the ICD-10 (Code: A01.0), hospitalised between 1 January 2010 and 30 September 2024. We collected patient data from the hospital’s electronic medical records, presented them as a case series and then analysed them. We concentrated on the demographic and epidemiological data, the history of the disease, the clinical and paraclinical data, the aetiological treatment and the evolution of the disease. The VBH is the only hospital in Romania that treats tropical diseases. Therefore, virtually all cases diagnosed with such a disease should be directed to our centre. According to the hospital’s internal procedures, suspected or confirmed cases are isolated until bacteriological sterilisation is demonstrated. The patient’s contacts are evaluated and monitored during the possible incubation period by the territorial epidemiological services to find possible secondary cases.

## 3. Results

During the 15-year period studied (January 2010–October 2024), we encountered a number of 13 cases diagnosed with typhoid fever and hospitalised in our centre. We did not find any case with a diagnosis of paratyphoid fever during this period. We compared the number of cases per year to the data reported by Romania to the ECDC, to the last available report in 2021 [13] and to the number of cases reported by our centre, the VBH, with a diagnosis of typhoid fever, which we show in Figure 1.

The cases described came from areas of high endemicity: Sub-Saharan Africa, the Indian subcontinent and Indonesia (Figure 2).

Based on the case history, we evaluated the periods between the onset of febrile illness and hospitalisation and compared them with the period between repatriation and hospitalisation, as shown in Figure 3.

The graphic in Figure 3 shows that the disease started in six out of nine patients before returning to Romania; only in three was the disease incubating at that time. The period from repatriation to hospitalisation was, on average, 13.2 days, with a median of 15 days, while the period from the onset of the disease to the time at which the bacteriological samples that established the diagnosis were collected was an average of 17.6 days, with a median of 15 days.

Figure 4 depicts the number of cases admitted to our hospital by year. In the COVID-19 pandemic years, there were no cases registered. In the last two years, six more cases have been described.

Below, we briefly describe the cases hospitalised from 2010 to 2024.

Case 1: A male patient, 32 years old, of Indian nationality, was hospitalised in February 2013 after a three-week stay in India and Nepal, four days after the onset of the disease. He did not present any significant medical history. The symptoms appeared 24 h after returning to Romania, with a high fever (40 °C), myalgia, cough, and headache. Clinically, he was an asthenic, febrile, tachycardic patient with moderate hepatomegaly. Laboratory: aneosinophilia, mild inflammatory syndrome (ESR = 26 mm/h), moderate ALT elevation (ALT = 109 U/L), mild renal impairment (creatinine = 1.3 mg/dL). Malaria and influenza were ruled out. Acute frontal sinusitis was diagnosed by clinical examination and radiography. The presence of *Salmonella* Typhi was demonstrated by blood culture, with urine and stool cultures being negative. The antibiogram performed did not reveal resistance to the tested antibiotics. He received treatment for 14 days with CIP 1 g/day. He became afebrile after the third day of antibiotic treatment, and the patient’s evolution was without complications. We assessed the patient as having a moderate form of the disease, and there was no evidence of the subsequent transmission of *S.* Typhi.

Case 2: Male patient, 57 years old, Romanian national, hospitalised after a three-month stay in India in September 2013. The disease started about 30 days before hospitalisation and evolved with diarrhoea, weight loss (more than 20 kg), fever, extreme asthenia, lack of appetite and a disseminated maculo-erythematous rash. During this period, the patient was not seen by a doctor and did not receive any treatment. Malaria, cholera and other bacterial aetiologies of the diarrheal syndrome were ruled out during hospitalisation. A complete blood count (CBC) showed lymphopenia (800/mm^3^), aneosinophilia and thrombocytopenia (108,000/mm^3^), moderate inflammatory syndrome (CRP = 4 mg/dL) and a biochemically moderate ALT elevation (ALT = 140 U/L). The presence of *S*. Typhi was evidenced in the blood culture and stool culture, and antibiograms showed resistance to CIP. He received CTX for eight days, to which we added chloramphenicol for four days; the evolution was slowly favourable, with the appearance of a non-febrile state after seven days. On the ninth day of hospitalisation, acute abdominal phenomena appeared, and intestinal perforation was diagnosed, for which reason he was transferred to a surgical clinic. The patient was subsequently lost from our clinic’s observation.

Case 3: A male patient, 22 years old, a Romanian national with no significant medical history, was hospitalised in May 2014 after a trip to Kenya. The illness started with fever, chills and sleepiness; the patient was hospitalised and diagnosed there on the fourth day of illness, by Widal test, with typhoid fever. Based on the anamnestic data, he was treated with CIP 1 g/day for five days, with good evolution. On his return to Romania, the patient was hospitalised and restarted treatment with CIP after a break of five days. On hospitalisation, he complained of slight asthenia without any other symptoms. The bacteriologic tests were negative, and the paraclinical data revealed no pathologic changes. We assessed the patient as having mild typhoid fever in convalescence, and there was no evidence of the subsequent carriage of *S*. Typhi.

Case 4: A female patient, 25 years old, Romanian national, presented on 20 October 2014, on the third day of a stay in India, with diarrheal stools and moderate fever, for which she received CIP 1 g/day and loperamide, with a slightly favourable evolution. She returned to Romania five days after onset and was seen by doctors in various clinics and received treatment with rifaximin and doxycycline (DO), because the patient had diarrhoea and a cough alongside a persistent moderate fever. On the 23rd day of the illness, the patient was hospitalised in our clinic due to persistent symptoms. The analysis revealed moderate anaemia, a low eosinophile count (3/mmc), no other hematologic changes and inflammatory syndrome (VSH = 87/mm^3^, CRP = 2.45 mg/dL). Microbiologically, although the presence of *S.* Typhi in the stool (blood cultures, stool cultures, urine cultures) could not be seen, *Shigella flexneri* and *Giardia lamblia* were present. The diagnosis of typhoid fever was confirmed by the Widal reaction, which was positive. The patient was treated with CTX for 12 days and albendazole for five days. The evolution of the patient was favourable; the diarrhoea syndrome disappeared after three days of treatment and fever after five days. This case was categorised as having a mild form of the disease, and we had no evidence of the subsequent carriage of the pathogen.

Case 5: A female 54-year-old patient with Romanian citizenship, who had been living in the Democratic Republic of Congo (DRC) for three years, presented with fever and back pain at the end of November 2015. She was admitted to a hospital in the DRC, where she was diagnosed with typhoid fever (positive Widal reaction). She received treatment with CIP, 1 g/day. She was hospitalised on the 11th day of evolution. She had been afebrile since the fifth day of treatment. On admission, she did not present any pathological changes. Laboratory findings: hemogram with normal values except for aneosinophilia, which increased from 600/mm^3^ to 900/mm^3^, without evidence of parasitosis or allergic diseases. She had no inflammatory syndrome or organic lesions evidenced clinically or by blood tests during admission. Bacteriological samples (urine, blood and stool cultures) were negative. She continued to receive treatment with CIP for another three days (total 14 days). The case was classified as moderate typhoid fever, in convalescence at the time of presentation.

Case 6: A female patient, a 24-year-old Romanian national who worked in India as a model between October 2017 and January 2018, presented with fever, chills and diarrhoea since 31 December 2017. She was repatriated four days after onset and presented to a general hospital due to worsening symptoms on the seventh day of illness. Here, anaemia, thrombocytopenia and hepatic cytolysis were found, which is why she was sent to our clinic with suspicion of malaria. Upon admission, the patient presented with an altered general condition; she was febrile, with signs of moderate dehydration, and had diarrheal stools. Malaria was ruled out upon admission. Blood tests revealed mild anaemia, lymphopenia (800/mm^3^), marked hypoeosinophilia (3/mm^3^), hepatocytolysis (ALT = 106 IU/L) and inflammatory syndrome (CRP = 16 mg/dL). A pulmonary infiltrative aspect with a bilateral pleural reaction was revealed, although the patient had no respiratory symptoms. *S.* Typhi was identified in stool and blood cultures as resistant to CIP and sensitive to the rest of the tested antibiotics. She received CTX 2 g/day for 12 days; the diarrheal stools disappeared on the fifth day of treatment and fever after eight days. She continued treatment at home for nine days with TMP-SMX. There was no evidence of the subsequent persistence of the bacteria. The case was considered an average form of the disease.

Case 7: A male patient, age 45, of Arab origin and a resident of Romania, was hospitalised for the investigation of typhoid fever diagnosed in India. He stayed in Mumbai for seven months, with the onset of the disease being at the beginning of October 2018, diagnosed via the Widal reaction. Symptoms at onset: fever, chills, myalgias, arthralgias. He received five days of treatment with cefixime (CFM), the evolution being favourable. He was admitted to our clinic 10 days after the onset of the disease. He was afebrile, with the only symptomatology being arthralgia in the left wrist joint, without clinical signs of arthritis. Blood tests revealed hypoeosinophilia (180/mm^3^), mild hepatocytolysis (ALT = 60 IU/L) and the absence of inflammatory syndrome. He received treatment with CFM for five days. The bacteriological samples collected were negative, and the patient had a positive Widal test. We assessed the case as having a mild form of the disease, convalescent at the time of admission to our clinic.

Case 8: A male patient, a 35-year-old Romanian national who was unvaccinated, who did not receive antimalarial prophylaxis and who had worked as a private soldier in the DRC since 1 April 2023, in the Goma region, presented on 23 June with fever, chills and headache and was repatriated to Romania on 1 July without undergoing any investigations or receiving any treatment. On 4 July, he developed diarrhoea; for this reason, two days later, he was admitted to a general hospital, and, on 8 July, he was admitted to our clinic. Upon admission: poor general condition, plateau fever (38 °C), headache, diarrhoea, abdominal pain. Laboratory tests revealed moderate leukopenia (3700/mm^3^) with lymphopenia (800/mm^3^) and aneosinophilia, hepatic cytolysis (ALT = 155 U/L) and inflammatory syndrome (CRP = 10.5 mg/dL). Evaluations for Zika, Chikungunya, Dengue or malaria viruses were negative. *S.* Typhi was identified in blood and stool cultures, and the pathogen was resistant to CIP, AMP and trimethoprim-sulfamethoxazole (TMP-SMX). He received treatment with CTX 2 g/day (10 days), AZM 500 mg/day (seven days) and dexamethasone 16 mg/day (seven days). Stools normalised on the fifth day and afebrility occurred after seven days of treatment. No subsequent transmission state was observed, and no secondary cases were observed.

Case 9: A male patient, a 27-year-old Romanian national, vaccinated against yellow fever and diphtheria–tetanus–poliomyelitis, worked as a military contractor in the Goma region, DRC, from March to June 2023. On 4 July, he was repatriated, and, three days later, he presented with fever, chills and the production of watery diarrheal stools lasting three days. Due to the persistence of the fever, he went to a general hospital in his city of residence on 14 July, where they found lymphopenia, hepatocytolysis syndrome and biological inflammatory syndrome. He received treatment with DO for three days, which did not influence the plateau fever; hence, he was sent to our clinic on 19 July. Blood tests performed upon admission revealed aneosinophilia without other changes in the CBC, hepatic cytolysis (ALT = 257 IU/L) and inflammatory syndrome (CRP = 11.8 mg/dL). Blood culture revealed *S*. Typhi resistant to AMP and CIP. He received CTX 2 g/day and AZM 500 mg/day for nine days. The fever disappeared after five days of treatment, and the patient was discharged on the 10th day of hospitalisation; this case was considered to have an average form of the disease, without complications.

Case 10: A male patient, a 48-year-old Romanian national, vaccinated against yellow fever, who worked as a professional soldier in Goma, DRC, for 4 months and who, in July 2023, was treated for 15 days with CIP 1 g/day for typhoid fever, with negative control stool cultures at the end; he was repatriated on 1 August. On August 10, he presented with fever (39 °C), chills, diarrheal stools and colicky abdominal pain and had self-medicated for 2 days with CIP. He visited a regional infectious disease hospital, which sent him to our clinic on 14 August. Investigations upon hospitalisation showed aneosinophilia, without other changes in the hemogram, as well as inflammatory syndrome (CRP = 10.9 mg/dL), increased LDH values (285 U/L) and hepatocytolysis. *S.* Typhi was isolated from the stool and blood cultures, and the antibiogram demonstrated resistance to AMP, CIP and TMP-SMX. Imaging did not reveal hepato-splenomegaly. Malaria, Dengue and Chikungunya were ruled out. He received treatment with CTX 2 g/day (14 days) and (AZM) 500 mg/day (10 days); the appearance of the stools normalised after three days, and the fever disappeared on the fifth day. The patient was discharged clinically cured, without evidence of subsequent pathogen transmission. We assessed the case as having a relapse of typhoid fever, with an average form of the disease.

Case 11: A female patient, a 29-year-old Romanian national, during a stay in Bali, developed diarrhoea lasting three days, headache, chills, high fever and vomiting episodes, as of 20th September 2023. For this reason, on the 26th, she was hospitalised for four days in a hospital in Bali with a diagnosis of typhoid fever (positive Widal test). For this, she received an injectable treatment with CTX for five days and subsequently oral CFM upon discharge. On 2nd October, she was repatriated, and she was hospitalised in our clinic on 8th October due to a persistent frontal headache, marked asthenia and coldness of the extremities in the absence of fever. Investigations performed revealed leukocytosis (19,000/mm^3^) with neutrophilia (14,600/mm^3^), acute inflammatory syndrome (CRP = 0.9 mg/dL) and an increased value of LDH (277 mg/dL) without hepatic or muscle cytolysis. The bacteriological samples collected (blood cultures, stool cultures, urine cultures) were negative. We performed serology tests for Salmonella Typhi (IgM, IgG), which were negative. She received treatment with CTX 2 g/day and AZM (500 mg/day) for nine days, with the remission of all symptoms at discharge. She remained afebrile throughout the entire hospitalisation period. The case was considered to be an average form, being clinically and bacteriologically cured at the time of hospitalisation in our clinic, and it could not be considered a relapse of the disease. However, there was no pertinent explanation for the symptoms of leukocytosis and neutrophilia that she presented when she was hospitalised in our clinic.

Case 12: A male patient, a 38-year-old Romanian national, without any prophylaxis related to pathologies endemic to the respective region, who was on a tourist trip during the period 21 February 2024–01 March 2024 in Sri Lanka and on 15 March 2024–17 March 2024 in Istanbul, Turkey. On 13 March 2024, he had a fever that progressively increased to 40 °C, chills and a cough. During his stay in Turkey, the patient presented with respiratory symptoms and received symptomatic treatment. Two days after the second repatriation, he arrived at an infectious disease hospital, where he received, for the respiratory infection, DO and Oseltamivir, with the evolution being unfavourable. On 21 March, we hospitalised him in our clinic, given the persistence of the described symptoms. The CBC upon admission revealed lymphopenia (500/mm^3^), aneosinophilia and thrombocytopenia (113,000/mm^3^), inflammatory syndrome (CRP = 15 mg/dL) and muscle cytolysis syndrome (creatine kinase = 436 U/L, LDH = 256 U/L). He presented with hepatocytolysis syndrome, with ALT that transiently increased during hospitalisation to 762 U/L. SARS-CoV-2, influenza, malaria, Dengue and Chikungunya were ruled out. The blood culture collected upon admission was positive, and the antibiogram demonstrated resistance to CIP. He received treatment with amoxicillin/clavulanic acid and DO for three days. Subsequently, he received CTX 2 g/day for 10 days and levofloxacin 750 mg/day for four days; we then changed to AZM 500 mg/day orally for seven days. Defervescence was obtained only after 48 h of treatment with AZM and after nine days of antibiotic therapy, although the administered beta-lactams were active against the pathogen. This can be explained by the inability of these antibiotics to act intracellularly. The case was considered a moderate form of the disease, with significant hepatic and respiratory damage.

Case 13: A male patient, 48 years old, of Indian nationality, who had been in Romania for two weeks as a worker, was admitted on 22 May 2024 by transfer from a general hospital due to fever, chills, headache and asthenia. The condition’s onset occurred five days earlier, with progressively worsening symptoms. The patient had an episode of typhoid fever in India (positive Widal reaction on 15 April 2024), for which he received antibiotic treatment that we could not document. Upon admission, the patient presented with fever (38.5 °C), aneosinophilia, inflammatory syndrome (CRP = 5.8 mg/dL) and mild hepatocytolysis (ALT = 60 U/L). We obtained the diagnosis through positive blood culture, and the antibiogram showed resistance to CIP. He received treatment with CTX 2 g/day for 12 days, with bacteriological control samples being negative. The case was considered to be an average form of the disease with relapse.

Table 1 shows the main characteristics of the patients at admission in our hospital.

Summarising the above data, we found that the symptoms at onset included fever (all cases), chills (seven cases), asthenia (six cases), headache (five cases), cough (three cases), myalgia and abdominal pain (two cases).

Table 2 presents the clinical and laboratory features of the patients diagnosed with typhoid fever in the VBH. We excluded the four cases diagnosed and treated in the countries where the disease appeared.

Of the nine cases diagnosed at the VBH, the aetiological diagnosis was established in eight by blood cultures; the stool cultures were also positive in two of them. One case was diagnosed with a positive Widal reaction. Susceptibility testing (antibiogram) was performed when the blood cultures were positive. In the presented case series, in 2013, we found the only case with *Salmonella* Typhi susceptible to all antibiotics. All other cases had resistance to CIP, and we found triple resistance for the 2023 cases coming from DRC (AMP, CIP, cefotaxime (CTX)). Except for case 2, which had severe complications, the forms of the disease were average, with defervescence appearing within three to seven days. The median number of days of hospitalisation was 10. In nine cases, the patients received treatment with CTX; this was combined with AZM in five cases and chloramphenicol in one case. In the other situations, three patients received therapy with CIP and one with CFM.

None of the patients became carriers, with all of them being cured at the end of the appropriate follow-up period. Only one patient developed a severe complication, namely peritonitis after intestinal perforation, but, in his case, the outcome was favourable as well.

## 4. Discussion

In this research conducted over 15 years, we found that, in specific years during this period (2010–2012, 2016, 2017, 2019–2022), we received no cases of typhoid fever. Except for 2023, when there were four cases, of which three came from the same region (Goma in the DRC), we did not encounter more than two cases yearly. Of the thirteen cases presented, the diagnosis of typhoid fever was obtained in the VBH in nine cases, with the rest being diagnosed and treated in the countries where the disease occurred. According to Figure 1, between 2010 and 2021, there were differences between the two reports (ECDC and VBH). In 2015 and 2016, more cases were reported nationally than in our hospital’s report. In 2014, when no case had been reported to the ECDC, we had one case at the VBH, diagnosed and treated in Kenya, which was only confirmed retrospectively in our clinic (case 3). The same situation occurred in 2018, when an additional case was reported in the VBH; compared to the national reporting, a patient diagnosed and treated in India was practically hospitalised during the convalescence period.

From the epidemiological investigations, we noted that none of the patients had been vaccinated against typhoid fever. In addition, none of the patients had received a prior travel medicine consultation. The possible consequence of this finding was a delay in presenting to the doctor after the onset of symptoms. This delay is shown in Figure 3, where we highlight the difference in the number of days between the repatriation and onset hospitalisations for the nine cases diagnosed in the VBH.

Delays in establishing a diagnosis are not only characteristic of “imported” cases of typhoid fever. Thus, in a study conducted by a hospital in the UK, they highlighted a median value of eight days [14]. In a study conducted in a country with high endemicity (Pakistan), they reported that the duration of the disease between the onset of fever and hospitalisation varied between five and 28 (median 20) days for patients with XDR typhoid and five and 30 (median 13) days for those with non-XDR typhoid (P < 0.001) [15]. However, it is known that faster antibiotic treatment decreases the percentage of complications [16]. A meta-analysis published in 2019 by Cruz Espinoza LM et al. [17] showed that patients hospitalised after 10 days of onset developed severe signs of disease 36% more frequently and had a threefold higher risk (OR, 3.00 [95% CI, 2.14–4.17]; P < 0. 0001) compared to those admitted earlier, where the percentage was 16%. Regarding the patients presented in this case series, five out of nine cases had a duration of more than 10 days, and only in one (case 2) did we find severe complications. However, although no secondary cases were diagnosed, extended periods of contact in the community could have constituted an essential risk of transmission, with one of the cases even having a 3-day stay in Istanbul during the disease state (case 12). Interestingly, in the context of the nine patients diagnosed in our hospital, the suspicion of typhoid fever in the referral diagnoses was only reported in the two cases (9 and 13) that had previously had a positive Widal test. In the other cases, the patients had a “febrile syndrome” as a referral diagnosis, or there was suspicion of malaria.

The characteristics of the patients diagnosed in the VBH at admission showed that most of them presented tachycardia, with a median of 106 beats/min, an unusual manifestation in the presentation of this disease. This could explain the fact that the patients presented acute myocarditis, a phenomenon that sometimes manifests as a complication of the disease in weeks 2–3, which is when these patients arrived at our clinic [18,19]. In eight of the nine cases, the patients had aneosinophilia, while lymphopenia, another characteristic of the disease, was seen in a smaller proportion (five of nine). Liver injury was seen in most cases (eight out of nine cases). In five out of nine patients, we observed pulmonary injury in the context of the disease, considered by previous authors as the “pulmonary signature” of typhoid fever. Only one patient presented a cutaneous injury (rose spot). Five out of nine patients received antibiotic treatments before hospitalisation, but their effectiveness was questionable in those who received CIP, with resistance to this antibiotic being demonstrated later. The antibiotic resistance of *S.* Typhi was shown, except in a single case from 2013, with MDR *Salmonella* Typhi, a characteristic recently encountered in many existing studies [8,9,10]. None of the strains presented resistance to CTX.

A particularity encountered in the three cases from 2023 from the DRC was that the three patients worked as private soldiers in the Goma region. The illnesses occurred in the same period (June–July 2023), without an epidemiological link between the cases. At that time, the World Health Organization reported a triple epidemic in this country of typhoid fever, shigellosis and cholera [20]. One of the patients had presented symptoms in the DRC, had not received medical care there and had been repatriated (case 8). Another (case 9) had left the DRC during the incubation period of the disease and had not received treatment until hospitalisation, and the third (case 10) had received treatment with CIP in the DRC without any further investigations other than a Widal reaction. All three had infections with *S.* Typhi MDR (CIP, AMP, TMP-SMX).

According to our presented experience, any patient with the following features should be suspected of having typhoid fever.***Fever*** (over 38.5 °C) lasting more than five days○and/or any of the following: diarrhoea, abdominal pain, constipation, anorexia or relative bradycardia.***Travel history*** in the past 21 days in an endemic area○primarily in the Indian subcontinent or sub-Saharan Africa;○especially if not vaccinated for typhoid fever or without a prior travel medicine consultation.***Low eosinophil count*** (up to aneosinophilia).Negative tests for malaria and Dengue fever.

According to the VBH’s internal procedures, blood cultures are collected from all febrile patients coming from endemic areas and, if appropriate, stool cultures, regardless of the suspected aetiology.

## 5. Conclusions

Although the disease is still endemic in many parts of the world, in countries with adequate sanitation, typhoid fever has become a “forgotten” disease due to the low incidence of cases, which makes it difficult to recognise by medical systems. Very often, when faced with a case of febrile syndrome originating from Southeast Asia or Africa, many doctors think first of vector-borne diseases (malaria or various viral diseases), with the presence of typhoid fever being suspected many times only in the context of the existence of diarrhoea, which is not a prominent symptom of the disease. This is why, in the previous section, we propose a set of criteria that may be considered as an algorithm to indicate the need for testing for typhoid fever in non-endemic countries, such as Romania. To our knowledge, this paper is the only one from the last decade that has addressed this topic in Romania.

## Figures and Tables

**Figure 1 idr-17-00016-f001:**
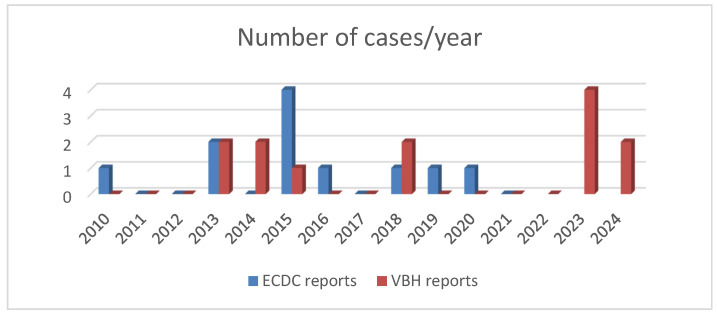
Number of cases/year according to the two reports (European Centre for Disease Prevention and Control, Dr. Victor Babes Hospital for Infectious and Tropical Diseases).

**Figure 2 idr-17-00016-f002:**
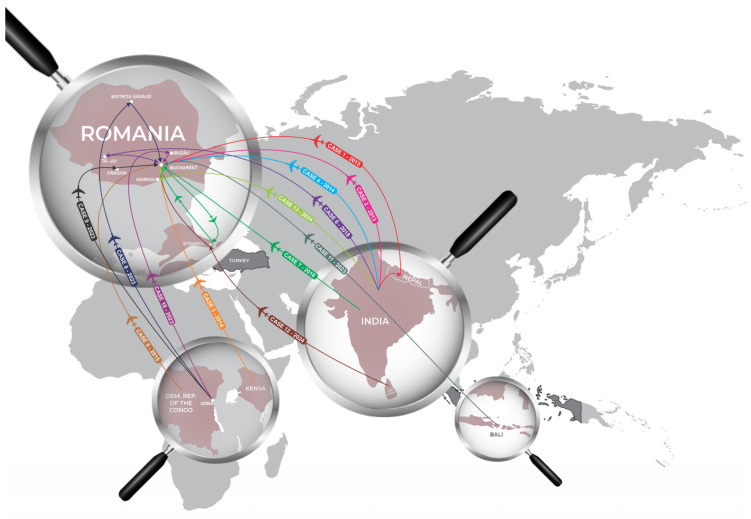
Origin and trajectory of typhoid fever cases admitted to Dr. Victor Babes Hospital for Infectious and Tropical Diseases during 2013–2024. Each color represents a different case.

**Figure 3 idr-17-00016-f003:**
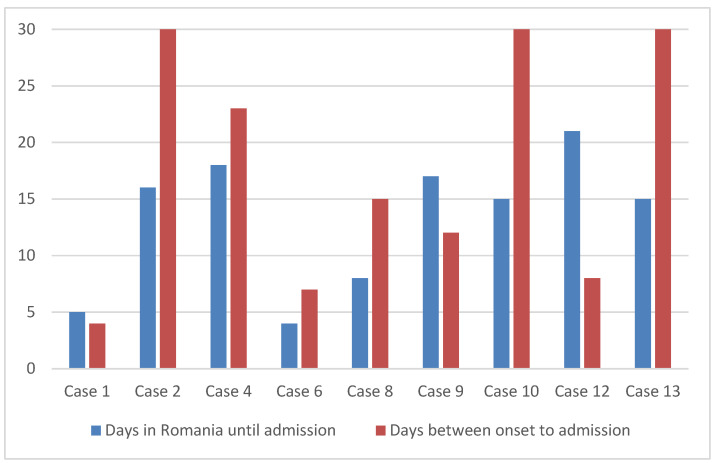
Comparison (in days) between the repatriation and onset hospitalisation periods.

**Figure 4 idr-17-00016-f004:**
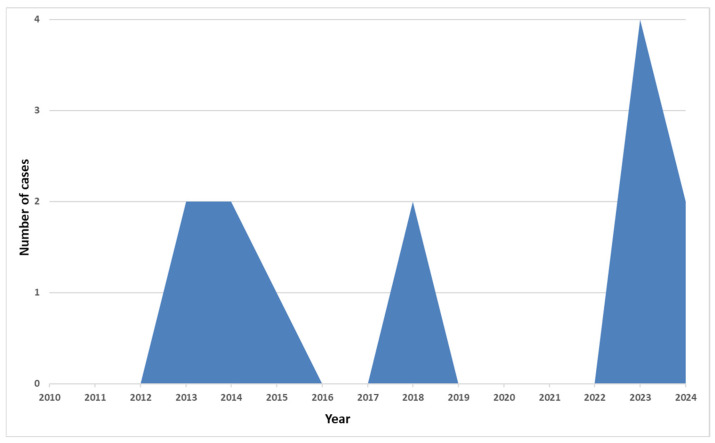
EPI curve of typhoid fever cases admitted to VBH.

**Table 1 idr-17-00016-t001:** Main characteristics of all patients diagnosed with typhoid fever.

	Case 1	Case 2	Case 3	Case 4	Case 5	Case 6	Case 7	Case 8	Case 9	Case 10	Case 11	Case 12	Case 13
Hospital admission	Feb-13	Oct-13	May-14	Oct-14	Nov-15	Jan-18	Oct-18	Jul-23	Jul-23	Aug-23	Sep-23	Mar-24	May-23
First symptoms to admission (days)	6	31	2	23	11	11	8	15	12	30	6	8	30
Method of diagnosis	blood culture	blood culture	Widal	Widal	Widal	stool and blood cultures	Widal	stool cultures	blood cultures	blood cultures	Widal	blood cultures	blood cultures
Place of diagnosis	VBH	VBH	Kenya	VBH	DRC	VBH	India	VBH	VBH	VBH	Bali	VBH	VBH
Typhoid fever vaccination	no	no	no	no	no	no	no	no	no	no	no	no	no
Other prophylaxis	no	no	no	no	yes	no	no	no	yes	no	yes	no	no
Country	India/Nepal	India	Kenya	India	DRC	India	India	DRC	DRC	DRC	Bali	Sri Lanka	India
Previous antibiotic treatment	no	no	CIP	CIP, DO, RFX	CIP	no	CFM	no	DO	CIP	CTX, CFM	DO	no
Hospitalisation (days)	15	9	7	14	14	14	6	10	10	10	10	14	12
Treatment	CIP	CTX + CHL	CIP	CTX	CIP	CTX	CFM	CTX	CTX + AZM	CTX + AZM	CTX + AZM	CTX + AZM	CTX
Atb. treatment (days)	14	9	5 + 5	12	14	12	10	10	9	14	9	10	12
Days until non-febrile state	3	7	0	5	5	8	0	7	5	5	0	7	5
Complications	no	intestinal perforation	no	no	no	no	no	no	no	no	no	no	no
Outcome	cured	unknown	cured	cured	cured	cured	cured	cured	cured	cured	cured	cured	cured

AZM—Azithromycin, CFM—Cefixime, CTX—Ceftriaxone, CHL—Chloramphenicol, CIP—Ciprofloxacin, DO—Doxycycline, RFX—Rifaximin,.

**Table 2 idr-17-00016-t002:** Characteristics at admission of patients diagnosed in VBH with typhoid fever.

	Case 1	Case 2	Case 4	Case 6	Case 8	Case 9	Case 10	Case 12	Case 13
Heart rate (/min)	100	106	80	80	100	112	117	114	162
Blood pressure (mmHg)	100/60	90/60	115/70	110/60	110/50	110/70	140/70	121/87	186/64
Anaemia	yes	no	yes	yes	no	no	no	yes	no
Lymphopenia	yes	yes	no	yes	no	no	yes	yes	no
Aneosinophilia	yes	yes	yes	yes	no	yes	yes	yes	yes
Low platelet count	no	yes	no	no	no	no	no	yes	no
Hepatic impairment	yes	yes	no	yes	yes	yes	yes	yes	yes
Renal impairment	yes	no	no	no	no	no	no	no	no
Respiratory impairment	yes	yes	yes	no	no	yes	no	yes	no

## Data Availability

The corresponding author can provide the data used in this study upon reasonable request.

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
