# Peer review of "Imported Typhoid Fever in Romania Between 2010 and 2024"

_2036-7449, 2025, doi:10.3390/idr17020016_

Round 1

Reviewer 1 Report

Comments and Suggestions for Authors

Lazar et al describe single cases of imported typhoid fever from various countries to Romania.

This is an important description of a neglected disease and might be important for general doctors (GP) in the country as it may take 5 – 20 days until admission and hence recognition.

The cases are well described.

What might surprise is the lack of infections of close contacts described in this report considering the long-time span between onset of the disease of 5 – 30 days until admission.

What might be of further interest for GPs is the precautions that are taken in Dr. Victor Babes Clinical Hospital for Infectious and Tropical Diseases to prevent spread of typhoid fever.

Author Response

Comments 1. Lazar et al describe single cases of imported typhoid fever from various countries to Romania.

This is an important description of a neglected disease and might be important for general doctors (GP) in the country as it may take 5 – 20 days until admission and hence recognition.

The cases are well described.

Answer 1. Thank you very much for your appreciation!

Comments 2: What might surprise is the lack of infections of close contacts described in this report considering the long-time span between onset of the disease of 5 – 30 days until admission.

Answer 2: You are right, this observation seems surprising, but we believe that good sanitation is a good way to prevent the spread of disease. As is known, typhoid fever is one of the diseases that "feel good" in areas where there is no proper water management in the community.

Comments 3: What might be of further interest for GPs is the precautions that are taken in Dr. Victor Babes Clinical Hospital for Infectious and Tropical Diseases to prevent spread of typhoid fever.

Answer 3: Thank you very much for your observation!. We added in the paper an explanation about the way to prevent the transmission of typhoid fever:

According to the hospital's internal procedures, suspected or confirmed cases are isolated until bacteriological sterilization is demonstrated. The patient's contacts are evaluated and monitored during the possible incubation period by the territorial epidemiological services to find possible secondary cases.

Reviewer 2 Report

Comments and Suggestions for Authors

In this manuscript, the authors present 13 case series of typhoid fever in Romania.

Due to growing population migrations (wars and other disasters, tourism, migrations for economic reasons), typhoid fever is an emergent disease that is a constant threat to the population, not only in endemic areas, but also in modern, highly developed countries.

However, several questions arise and require clarification.

First of all, according to the instructions for the authors, which are available on the journal's website, I am not sure whether this journal accepts the publication of case series.

Furthermore, the Abstract is unclear (lines 24-26) should be rewritten , and line 27 should be "all strains were sensitive to ceftriaxone.

The name Salmonella Typhi is not written correctly here, but also in several parts in the manuscript.

Introduction is poorly written and difficult to follow, even though it provides valuable information. All antibiotics should be uniformly written with a small letter. Line 27: MDR strains are those resistant to at least three antibiotics from three different antibiotic classes.  References for definitions of MDR and WDR are missing.

At the end of the introduction, "aim" is missing (lines 407-408 can be switched here).

Material and methods:

Line 89: "paraclinical"- please use another words (laboratory, clinical signs, microbiology...)

Results:

In my opinion, it would be much better if the clinical cases were presented in detail in the table (e.g. columns with the date of admission, date of the first symptoms - which symptoms, laboratory findings, how and when the diagnosis was made, whether the patient received prophylaxis, where they have traveled, how they were treated, complications,what the outcome was...). I think that this way of presentation would significantly improve the manuscript. The main features of individual cases would be much easier and faster compared, and the main conclusions of the research would be better presented.

In conclusion, the authors could present the distribution of the patients by severity of clinical presentation, how many of them became carriers, how many MDR isolates were present, how many cases were diagnosed by Salmonella cultivation and how many by Widal agglutination, and what the outcome was. Also, perhaps the authors could recommend a diagnostic algorithm for symptomatic travelers from endemic areas (based on the presented results).

The authors could create epi curves, it would be interesting to see the distribution by year.

Author Response

Comment 1: First of all, according to the instructions for the authors, which are available on the journal's website, I am not sure whether this journal accepts the publication of case series.

Answer 1: Given the small number of cases available, the publication was meant like an article based on a case series, but also analyzing the data available from ECDC and national authorities regarding typhoid fever in Romania. As mentioned, we are Romania's only tropical diseases hospital, and all cases are addressed to our unit.

Comment 2: Furthermore, the Abstract is unclear (lines 24-26) should be rewritten, and line 27 should be "all strains were sensitive to ceftriaxone.

Answer 2: Thank you for your suggestions! We have rewritten it according to your observations, and we hope that brings more clarity to the abstract.

Comment 3: The name Salmonella Typhi is not written correctly here, but also in several parts in the manuscript.

Answer 3: Thank you! We have corrected all the instances in the text.

Comment 4: Introduction is poorly written and difficult to follow, even though it provides valuable information. All antibiotics should be uniformly written with a small letter. Line 27: MDR strains are those resistant to at least three antibiotics from three different antibiotic classes.  References for definitions of MDR and WDR are missing.

Answer 4: Thank you for your observation! We have rephrased the Introduction chapter, and we hope it is easier to follow. We also corrected all the other issues you mentioned in your comments.

Comment5: At the end of the introduction, "aim" is missing (lines 407-408 can be switched here).

Answer 5: We have introduced the paper's purpose and the proposed topic's importance at the end of the introductory chapter. Thank you very much for the suggestion!

Comment6: Line 89: "paraclinical"- please use another words (laboratory, clinical signs, microbiology...)

Answer 6: We have changed the formulation. Thank you!

Comment 7: In my opinion, it would be much better if the clinical cases were presented in detail in the table (e.g. columns with the date of admission, date of the first symptoms - which symptoms, laboratory findings, how and when the diagnosis was made, whether the patient received prophylaxis, where they have traveled, how they were treated, complications,what the outcome was...). I think that this way of presentation would significantly improve the manuscript. The main features of individual cases would be much easier and faster compared, and the main conclusions of the research would be better presented.

Answer 7: Thank you for your suggestion! We have created a table with the main characteristics of all patients and inserted in the paper.

Comment 8: In conclusion, the authors could present the distribution of the patients by severity of clinical presentation, how many of them became carriers, how many MDR isolates were present, how many cases were diagnosed by Salmonella cultivation and how many by Widal agglutination, and what the outcome was. Also, perhaps the authors could recommend a diagnostic algorithm for symptomatic travelers from endemic areas (based on the presented results).

Answer 8: We added the Conclusion chapter and created a set of criteria that can be considered an algorithm for typhoid fever testing (can be found at the end of Discussions). We also addressed all other issues you mentioned in the Results chapter. Thank you!

Comment9: The authors could create epi curves, it would be interesting to see the distribution by year.

Answer 9: We added an EPI curve of cases per year in VBH, Figure 4. Thank you!

Round 2

Reviewer 2 Report

Comments and Suggestions for Authors

Dear authors, thank you for the opportunity to consider your revised manuscript.

You have successfully responded to all requirements, clarified all the issues and significantly improved the quality of the manuscript.

My opinion is that the manuscript is scientific interesting, clearly presented and brings significant results and conclusions.

It was a pleasure to read the new version of the manuscript.

Author Response

Dear Reviewer,

Thank you for your suggestions and indications that helped improve the manuscript! 

Thank you also for your appreciation!